# Evaluation of Chip Formation Mechanisms in the Turning of Sintered ZnO Electro-Ceramics

**Jaka Dugar, Awais Ikram *** and **Franci Pušavec ***

Faculty of Mechanical Engineering, University of Ljubljana, Aškerčeva cesta 6, SI-1000 Ljubljana, Slovenia; jaka.dugar@fs.uni-lj.si
* Correspondence: rana.awaisikram@yahoo.com (A.I.); franci.pusavec@fs.uni-lj.si (F.P.)

**Abstract:** The sintered zinc oxide (ZnO) electro-ceramics are a brittle class of hard-to-cut materials such that shaping them with the post-finishing operations necessitates careful handling and precision machining. The conventional machining approach using the grinding and lapping processes represents limited productivity, an inability to produce the required geometries and frequent uncontrolled chipping of the edges of the final products. This study thus investigates the turning performance of dense sintered ZnO varistors and chip formations to obtain the parametric range (cutting mechanism) which causes the chipping or the trans-granular/sudden failure in these brittle materials. With the analysis of the cutting tool vibration in relation to the machining parameters ($f$ and $V_C$), the vibration-induced chipping correlations are made and interlinked with the occurrence of grain pull-out during the turning operation. The results show that the reflected vibratory motion of the tools is directly correlated with the chip formation mechanisms in the turning of ZnO ceramics and thus provide robust measurements for quality assurance in final products.

**Keywords:** ZnO varistors; machining optimization; turning; chipping; accelerometer; FFT plotting; 3D topography

## 1. Introduction

The sintered ZnO electro-ceramics are a class of functional materials that are highly brittle due to low fracture toughness, so finishing them with the post-finishing operations necessitates careful handling and precision machining [1]. The ZnO varistors require post-sintering surface preparation or edge turning for the development of metallization layers (Al, Ag paste) to join electrical leads/interconnects and a glass glaze around the sides [2]. Lapping is considered as a contemporary industrial practice in the machining of brittle ZnO ceramics. The conventional machining approach for hard ZnO ceramics is non-productive as the grinding and lapping technologies represent limited productivity due to frequent uncontrolled chipping or a sharpening of the finished products' edges at the expense of high machining costs and tool blunting [3,4]. Moreover, the material removal rate is quite low which makes conventional machining significantly difficult with poor sustainability [1]. On the finished parts, these chipped edges are thus unfavorable for the functional and mechanical properties.

Therefore, as an alternative, the focus should return to the micro-machining (milling and turning) of varistors with defined cutting geometry (sharp and defined edges with R < 15 μm). However, it is important to understand the shear force-induced chipping behavior in ZnO sintered ceramics to optimize the machining parameters for concurrent industrial utility and requirement. The turnaround industrial waste of these ZnO ceramics failing in the sintering or electrical impulse testing above 12 kA accumulates to approx. 15% of the total production. A prime method to enhance the machineability of the brittle ceramics is the thermal activated turning operation on the workpiece, where heat is applied to induce localized softening in the regions to be machined [5–7].

The vibrations during the machining sequence can have a significant impact on the workpiece and the tool life as well as the possible failure. The vibration analysis can be utilized for predictive maintenance practices in machining and for the decisions on machinery/tooling maintenance. This kind of vibration analysis performed before the failure is helpful to avoids the catastrophic consequences, from work part and tooling to the machine unit itself. Essentially, monitoring the vibrations generated during the machining processes offers benefits such as an improvement of production efficiency and downtime, a reduction in maintenance costs, added availability of machinery for simultaneous tasks, and lessening the stocks of tooling and/or spare parts. Accelerometers are a kind of sensor connected to the machining units for real-time monitoring of the vibrations during an operational sequence by the vibration amplitude signals from the time domain to the frequency domain with the aid of Fast Fourier Transform (FFT) algorithms in designated waveform patterns. These can be examined for fault detection and surface integrity analysis during the machining run. The accelerometer has also found similar applications in the characterization and optimization of metal cutting in the turning processes [8]. Roy et al. [9] successfully employed an accelerometer coupled to a machining unit to corelate the tool–workpiece contact before commencing precision machining. Chen et al. [10] reported on the application of an accelerometer to acquire a forecast related to tool breakage. Huang et al. [11] indicated the suitability of MEMS-based accelerometer sensors for vibration and fault monitoring in the milling process and this approach can be expanded to chatter recognition within the machining tasks as well [12]. The usefulness of accelerometer-based vibrational signals in analyzing the tool wear condition has also been investigated during the high-speed machining (HSM) of a titanium (Ti-6Al-4 V) alloy [13]. Moreover, the vibrational feedback from the accelerometer can be used to simulate and model the cutting forces and tool tip behavior in the milling units [14]. The tooling condition monitoring is also beneficial in characterizing the end result of the machining processes and the classification of the surface roughness [15]. Usually, multi-sensor accelerometer units are proficient in supporting the milling and machining processes by rendering real-time vibrational feedback of the cutting forces and tooling characteristics [16]. Researchers have also incorporated soft computing and statistical methods with the accelerometer feedback data [17] to predict and model the surface roughness during the turning [18] and milling operations [19].

Since understanding the turning characteristics of ZnO electro-ceramics is important due to the industrial requirement to coat dielectric around the circular cross-section as well as the metallization layer deposition on the top and bottom, the geometric precision of the profiles is important for these different types of functional coatings. The literature to the best of our understanding and from an exhaustive review does not contain reports on the turning performance and the parametric range classification of ZnO sintered ceramics. Therefore, this study investigates the turning performance of dense sintered ZnO varistors to obtain the parametric range (cutting mechanism) which causes the chipping or the transgranular/sudden failure in these brittle materials that are predominantly due to the Bi-rich intergranular phase and the elimination of the current breakthrough from the edge of the varistors. Important parameters, e.g., the surface integrity/roughness, tool wear, chamfering radius, the vibration-induced chipping with respect to the cutting speed and the feed rate are classified in our results for a diamond-based ultrahard tool. The accelerometer sensor in parallel with turning supports data acquisition related to the range of vibration-induced chip formations and simultaneously, the measurement of the cutting forces was corelated for different functional materials. Preliminary results imply that for the milling parameters: depth of cut 0.1 mm and width of cut 0.33 mm, respectively, a feed velocity of 250 mm/min and a spindle speed of 6250 rev/min ($V_C$ = 78.5 m/min and *fz* = 0.04 mm) [1], varistor ceramics experienced pronounced average and maximum chipping widths (edge chipping). The feed rate optimization was also obtained from the preliminary results and the most significant range is presented in this manuscript for a reasonable comparison with ceramic surface topography, which directly

influences the metallization and dielectric deposition as well as the IV performance under transient impulse current testing. The surface and failure mode characterizations made with a precise three-dimensional microscope system and a scanning electron microscope infer directly that the machined surface roughness is interlinked with the occurrence of grain pull-out during the milling operation. This work correspondingly clarifies why conventional machining is not suitable due to edge chipping in ZnO varistors owing to their low fracture toughness of 2.16 MPa·m$^{0.5}$ which degrades further with poorer relative densities in sintered compacts, especially at the edges.

## 2. Experimental Methodology

The measured geometric specifications of the ZnO varistors are presented in Figure 1. They had a diameter of 42 mm, a thickness of 12.6 mm, an edge radius r > 110 μm, a surface roughness (average height of selected area) classified in ranges $S_a$ ~ 1–1.4 μm and the maximum height of the selected area from 10 averaged points by $S_{10z}$ = 25–40 μm.

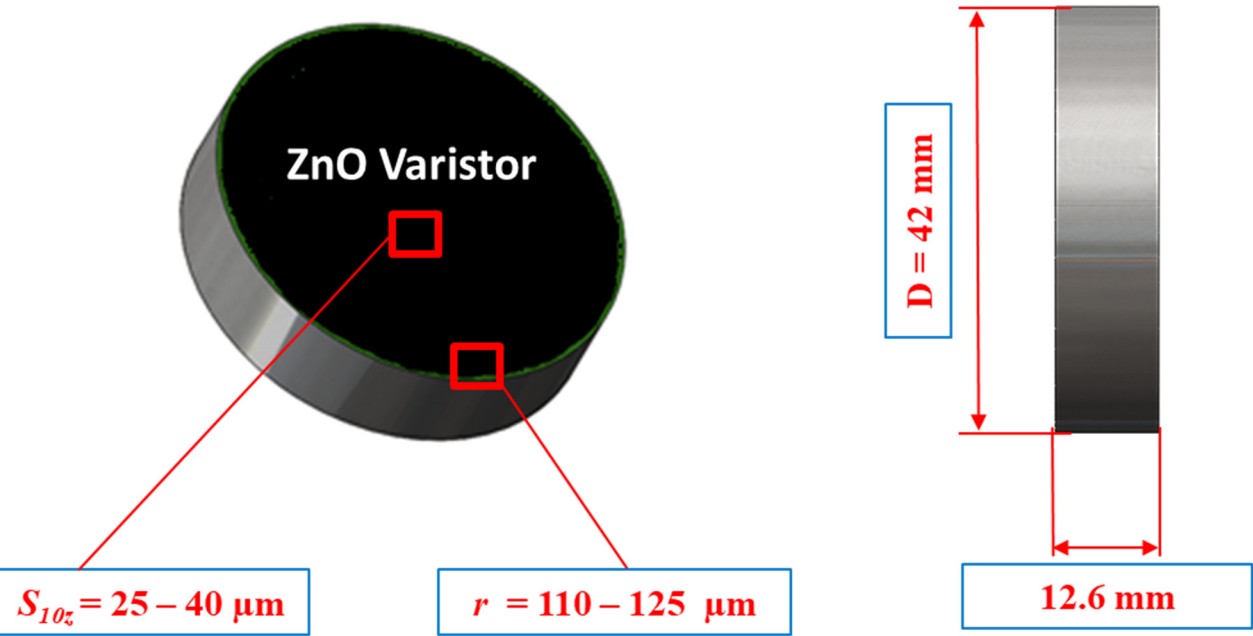

**Figure 1.** Geometric specifications of the varistor ceramics used in the turning operation.

The ZnO varistors were fabricated with the uniaxial pressing of the green compacts prior to the sintering operation and the two sides may have had different turning characteristics; therefore, color markings were introduced as shown in Figure 2 to identity the variation in the collection of the accelerometer signals related to surface roughness and material removal. The top side marked with the red color indicated the side which received the bulk of the compaction pressures during the uniaxial pressing which indicates a higher relative surface density on the top (red) part, whereas the blue (bottom) part indicated the fixation away from the pressure plungers. This illustrative scheme of the red and blue sides will be consecutively used in the results and discussion sections.

The turning operations were performed on a CNC Mori Seiki SL 153 lathe machine with a cutting polycrystalline diamond (PCD) type cutting tool (DCGW11T304) using the SDJCR/SDJCL–2020 cutting tip holder (illustrated in Figure 3), with a cutting edge angle κ inclined at 3° with the specimen surface and a cutting-edge radius of 0.4 mm. The following turning parameters were utilized to machine the surfaces of disk shaped varistors and to obtain acceleration feedback with the accelerometer during the turning process. The stud mounted Triaxial DeltaTron® Accelerometer Type 4525-B-001 was attached to the turning machine to record the vibrations in relation to the increase in the material feed rate

during the machining operations. The accelerometer was calibrated to record vibrational amplitudes in a range from 1.0 to 5.5 kHz for time periods of up to 14 s.

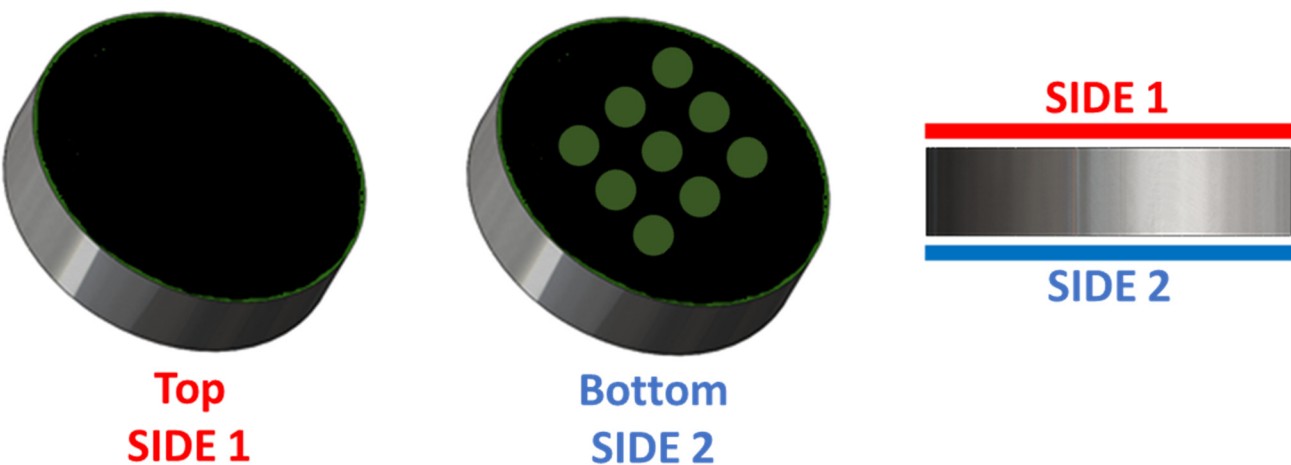

**Figure 2.** The division of the ZnO ceramics turning operations into two sides, illustrative of the uniaxial pressing of green compacts prior to the sintering operation.

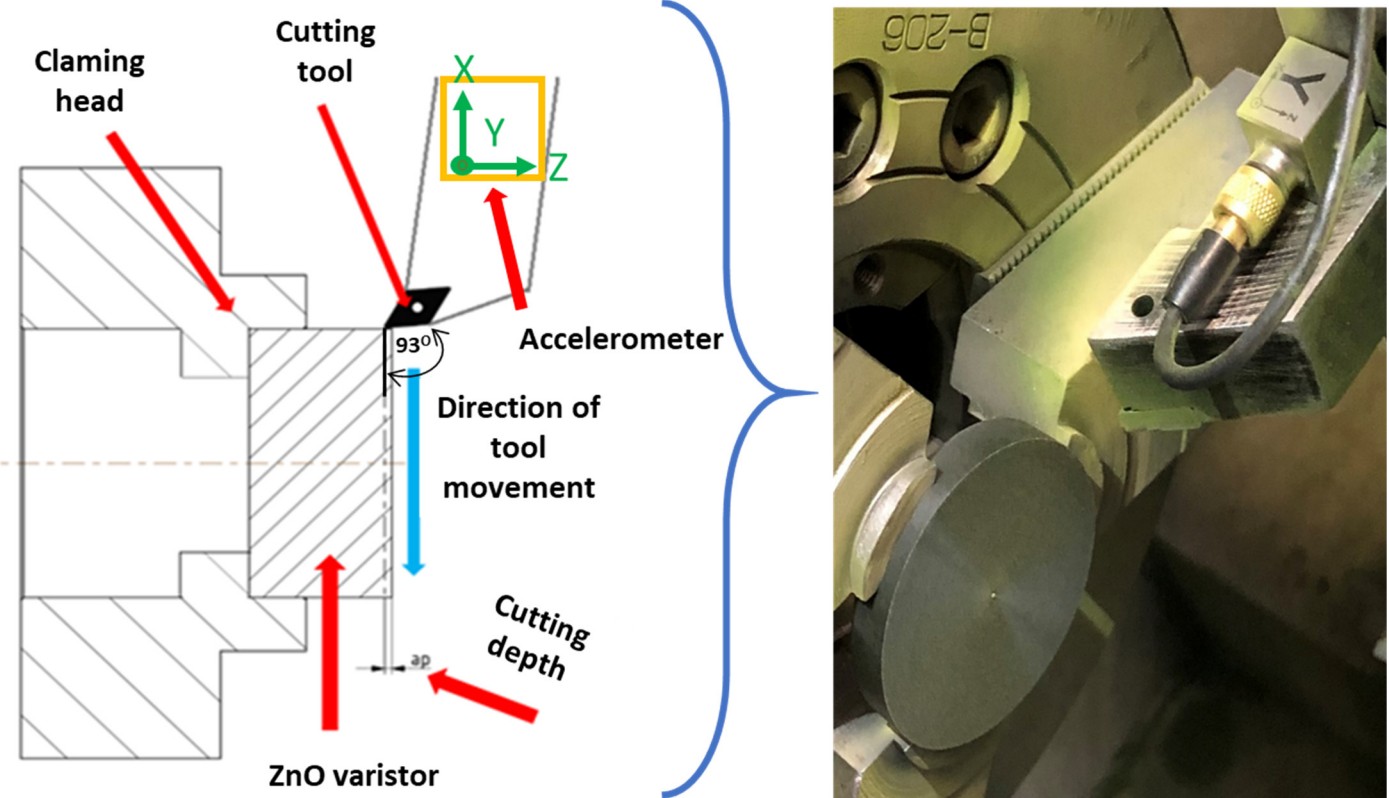

**Figure 3.** Schematics of the turning operation on a ZnO varistor.

The schematic of the turning operation is presented in Figure 3. The depth of cut ($a_p$) was maintained at 0.5 mm and the feed rate (feed velocity—$f_n$) was varied from 0.1 mm/rev and 0.15 mm/rev to 0.2 mm/rev, while the cutting speed ($V_C$) was kept at = 100 m/min and the spindle rotation speed (n) was kept at 3000 rpm, in-line with the preliminary studies with laser milling [1]. The surface integrity analysis was based on the surface roughness of the machined ceramics and the vibrational amplitude was based on the edge chipping

while the feed rate was varied. All turning experiments were performed in dry cutting conditions. All the ZnO workpieces used in the experiments were mounted and fixed to the same tightness on the turning unit.

The evaluation of the surface integrity was made with an Alicona InfiniteFocusSL measurement system in the form of three-dimensional scans of the surface and edges, generated in two passes. The topographical features were resolved with the Alicona three-dimensional optical microscope system utilizing the ISO 25178 method to compute the areal roughness and to calculate the $S_a$ and $S_{10z}$ parameters. Figure 4 illustrates the three-dimensional surface integrity setup with the ceramic workpiece mounted in the tilted position within the specimen holder and the top-down optical light illumination. The Alicona microscope takes advantage of the shallow depth of field of the optical system by vertical scanning (z-axis with precise piezoelectric positioning system) and obtains data on topography (in the form of a color scheme) by changing the focal length. The minimum measurable profile roughness $R_a$ at $10\times$ and $20\times$ magnification was 0.3 μm and 0.15 μm, respectively.

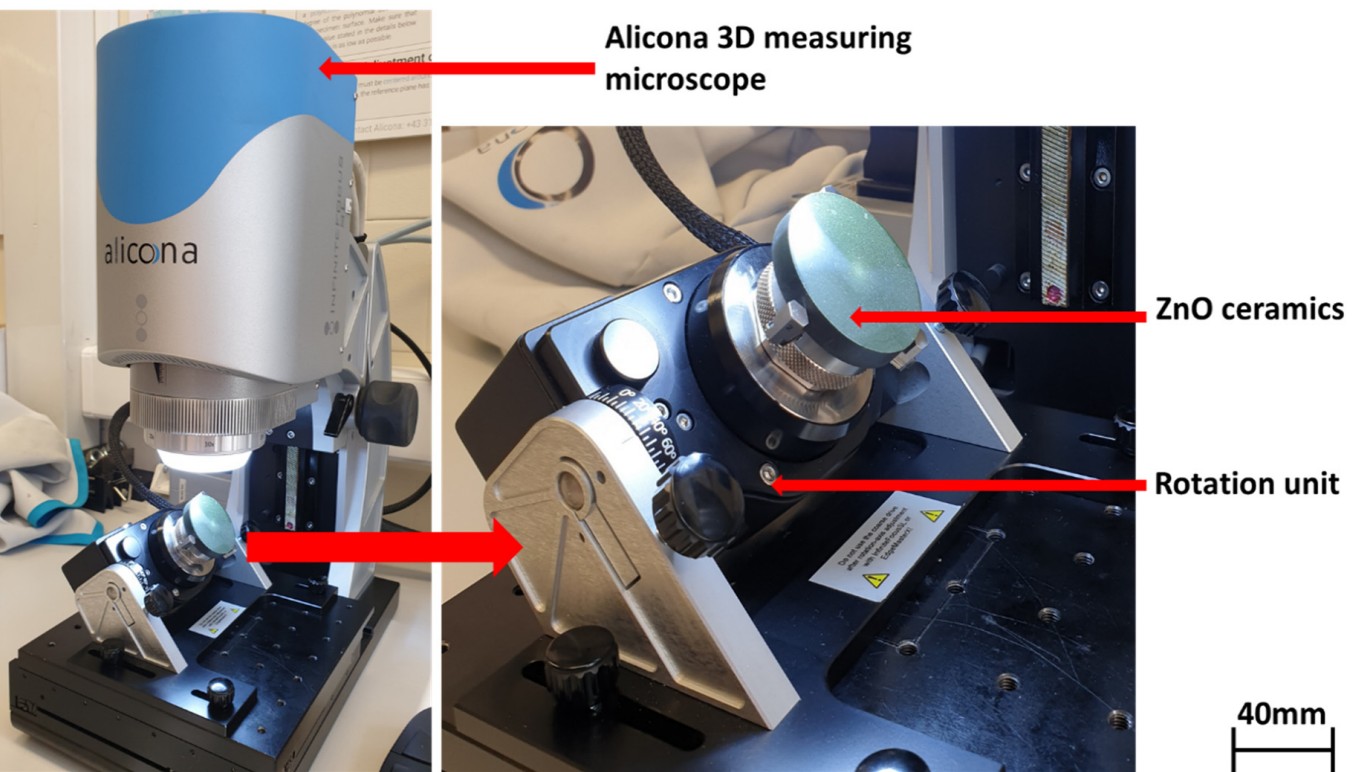

**Figure 4.** Surface roughness measurement setup with an Alicona 3D microscope.

The non-contact profilometry (edge) for the quantification of the profile roughness was measured by utilizing the ISO 4287 and ISO 11562 standards in the Alicona three-dimensional measurement system. The $R_z$ (average peak to depression height of the roughness profile) parameter was chosen to characterize the profile radius and roughness. These values, accurate to two decimal places, were returned by the software after the development of profile plots from 100 points along the specified dimension.

The surface roughness measurements under different feed velocities were measured on the top (marked red) and bottom (marked blue) sides of the machined ceramics, on randomly selected 5 mm long profiles in the x-axis direction. A repetition of the surface roughness measurements was considered along the y-axis approximately 10 μm apart to obtain the mean values.

The electron microscopy imaging was performed by a JEOL 7600F field emission scanning electron microscope at an accelerating voltage of 20 kV in secondary electron

mode to obtain the surface and topographical view of the ceramics, whereas the in focus backscattered electron detector at a working distance of 15 mm was utilized to obtain the phase contrast information. An EDXS point phase analysis was performed with an Oxford Inca 20 mm$^2$ detector utilizing 20 kV of accelerating voltage and a probe diameter of approximately 1 μm.

## 3. Results and Discussion

### 3.1. Accelerometer Turning Signals Feedback

The investigation of the vibration signals during the turning operation was performed with the accelerometer, to corelate the surface topography with the chipping frequency and the optimal machining characteristics of ZnO varistors. These accelerometer measurements accounted for the time and frequency domain FFT signals of the spectral power density obtained by the real-time vibrational feedback [20,21]. The FFT graphs presented in Figure 5 illustrate the time domain variation of the feed velocities exhibiting a higher signal attenuation from 0.1 mm/min to 0.2 mm/min, with measurements along the *x*-coordinate ($a_x$) showing the variation in the feed rate and the measurements along the *y*-coordinate ($a_y$) showing the variation in the cutting force in the form of vibrations. The results of the time domain FFT indicate that for a depth of ($a_p$) kept at 0.5 mm and in the case of a lower feed velocity of 0.1 mm/rev, the cutting operation was completed in slightly over 9 s. In contrast, the machining operation was performed at 6.3 and 5 s for 0.15 mm/rev and 0.2 mm/rev feed rates, respectively. Considering the $a_y$ segment of the time domain FFT plots, a similar result can be inferred: for a higher feed rate there is a proportional relationship of force feedback vibrations with the time series as the vibrations are higher for a 0.2 mm/rev feed rate. However, the difference with 0.15 mm/rev was insignificant. On the contrary, an inverse linear relation exists between the cutting force and the vibrational attenuation of the force feedback with time series: a higher cutting force resulted in a reduced time to complete the turning task. The vibrational feedback obtained from 0.2 mm/rev signifying the higher cutting force in play is quite evident in Figure 5; however, the requirement for the cutting force was lower for the higher feed velocities in the time domain to complete the respective turning operation on a sintered ZnO ceramic. In short, the turning operations finished faster with higher feed velocities and the process was completed in a shorter time frame at 0.2 mm/rev as compared to 0.1 mm/rev, which is in line with the projections from the accelerometer data related to machining processes [8,11,17].

Likewise, for the top side of the ZnO varistor where uniaxial pressure was applied prior to sintering, the frequency domain FFT plot is illustrated in Figure 6. The frequency response designates the maximum possible deviation of sensitivity over the frequency range, which was set at 12 kHz for the turning operations. Typically, the frequency response generated at the high frequency amplitudes primarily denotes the mechanical resonance of the sensor. The amplitude along the *x*-coordinates shows the variation in the feed velocities, while the amplitude signal produced along the *y*-coordinates refers to the variation in the cutting forces. In the case of the turning operations along the *y*-coordinates, the incremental rise in the vibrational amplitudes with the feed rates suggests higher cutting forces from 0.1 mm/rev to 0.2 mm/rev. As can be seen in Figure 5, the vibrational signal from the edge regions was small at a low feed velocity (0.1 mm/rev) between the frequency range of 7000 and 9000 Hz along the *y*-coordinates, whereas it became momentous when the feed rate was increased from 0.15 mm/rev to 0.2 mm/rev. The vibrational attenuation in the frequency domain was significantly pronounced at the central part of the ceramic as compared to the edges and the amplitude signals suggest significantly high cutting forces applicable in the frequency range from 9000 to 11,500 Hz on the surface section of the ceramic at 0.2 mm/rev feed velocities.

Similarly, the acceleration signal during the turning operation under different feed velocities of the bottom part of the ZnO ceramic is shown in Figure 7. The subsequent patterns of the acceleration signals $a_x$ (feed rate variation) and $a_y$ (cutting force variation), respectively along the (*x,y*)-coordinates for the bottom side of ceramic were similar to the

results obtained in Figure 5. Synonymously, with the depth of cut ($a_p$) retained at 0.5 mm, the cutting speed ($V_C$) = 100 m/min and the spindle rotation speed (n) at 3000 rpm, the turning operations in the time domain were completed more rapidly for the 0.2 mm/rev velocity as compared to the lower feed rates. The turning operations were completed in just over 9 and 6 s for 0.1 mm/rev and 0.15 mm/rev respectively, whereas they were completed more rapidly in under 5 s for 0.2 mm/rev. This indicates a slightly faster machining operation for the bottom side in the time domain as compared to the top surface. The magnitude of the acceleration signals generically gives qualitative feedback, indicating lower applied cutting forces in this case than the top surface, in line with the forecast models [16,18,20,21].

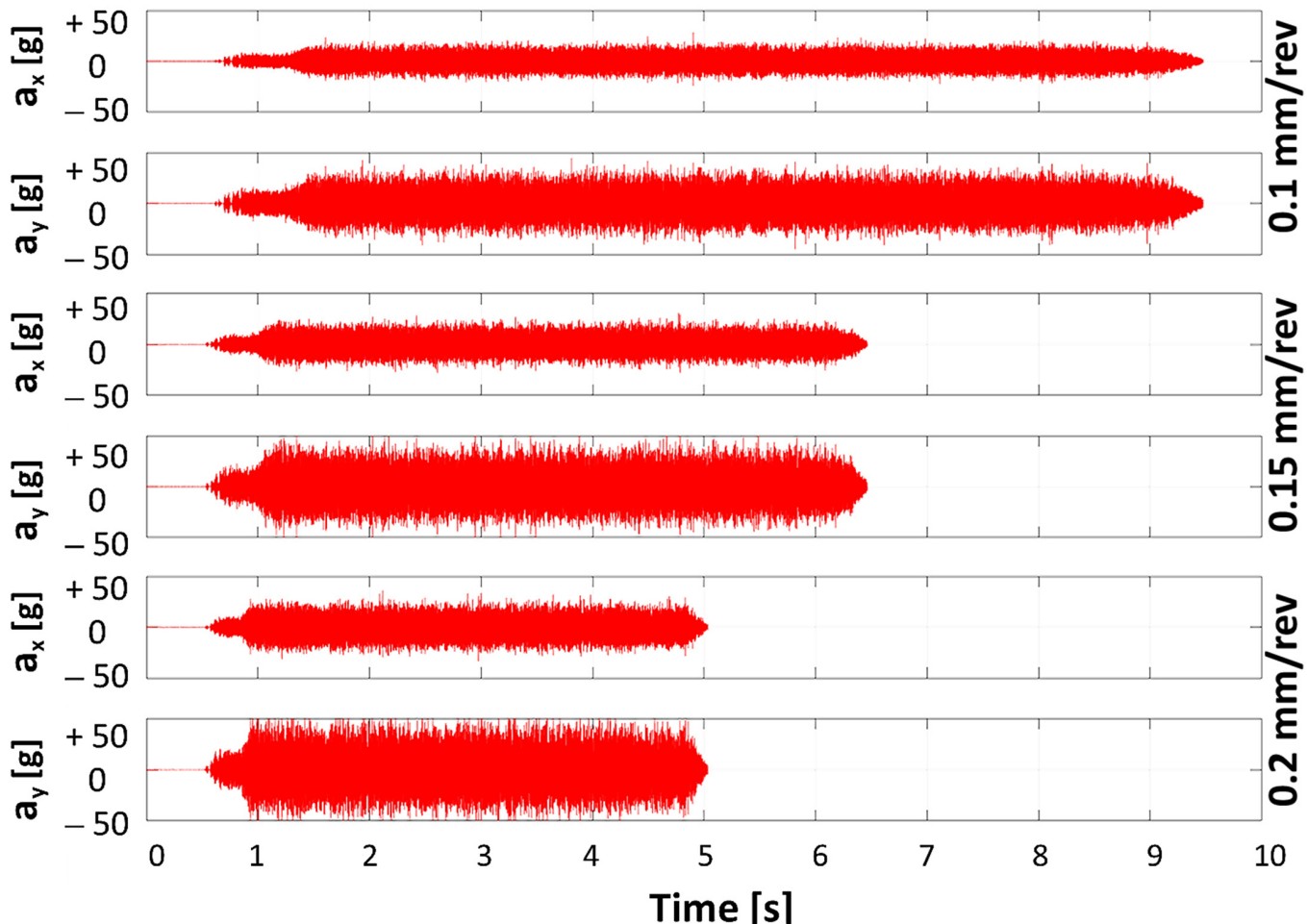

**Figure 5.** FFT plots of the acceleration signal in *x* and *y* coordinates with respect to time for the top side of the ZnO ceramic under different feed velocities.

The amplitude FFT graph for the bottom part is presented in Figure 8 for the *x* and *y* coordinates with respect to the different feed velocities. The two generic differences observed in the frequency domain illustration of the turning operations on the bottom surfaces are firstly, the lack of vibrational signals at the edges in the range from 7000 to 9000 Hz and secondly, the amplitude feedback (Amp Y—cutting force variation) during the machining for the 0.15 mm/rev and 0.2 mm/rev feed rates in the central segments of the sintered ceramics were slightly on the higher side. The spread of the frequency signals for 0.2 mm/rev was also broader for the bottom side, which implies that higher applied cutting forces result in shorter cutting times on higher feed rates.

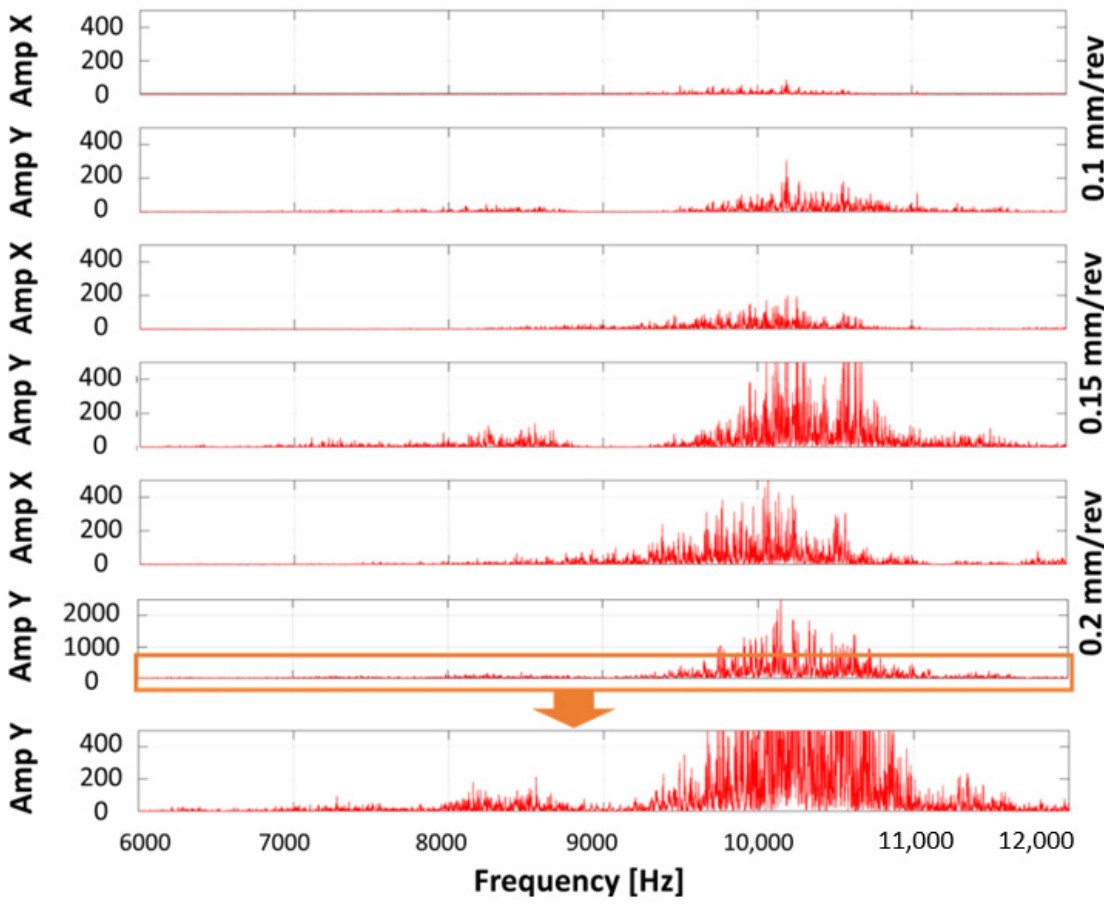

**Figure 6.** Frequency domain FFT plots in *x* and *y* coordinates for the top side of the ZnO varistors turned with different feed velocities.

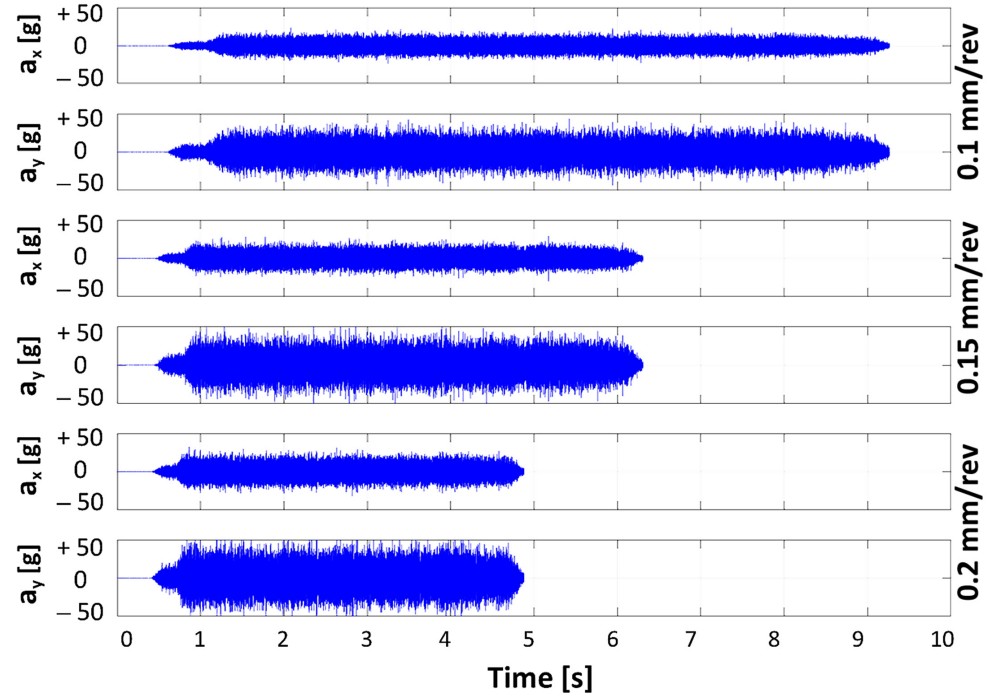

**Figure 7.** FFT plots of the acceleration signals in *x* and *y* coordinates with respect to time for the bottom side of the ZnO ceramic under different feed velocities.

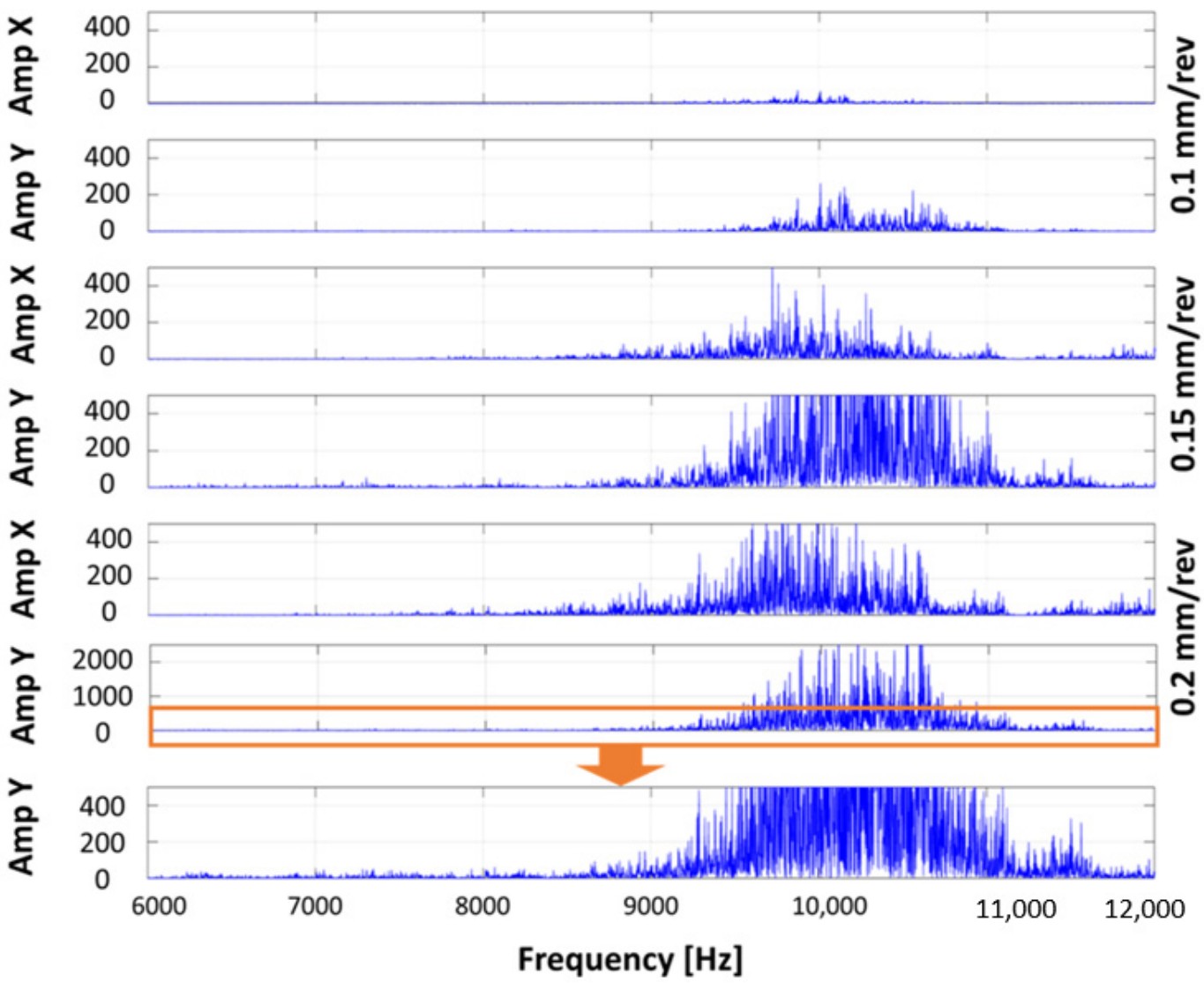

**Figure 8.** The FFT plots of the frequency domain in the *x* and *y* coordinate for the bottom side of the ZnO ceramic under different feed velocities.

This difference in the acceleration and the frequency FFT signals for the top and bottom parts further required microscopical characterization to determine the plausible changes in the machined surfaces as a result of the variation in feed rates, which is discussed in the ensuing section.

### 3.2. Electron Microscopy Characterization of the Machined Surfaces

Metal oxide varistors (MOV) are voltage-dependent resistors (VDR) which have non-linear, non-ohmic current–voltage characteristics to provide protection against the voltage transients and high current surges [22]. The non-linear characteristics are strongly dependent on the microstructure, which in-turn is related to the chemical composition and the applied processing methods [23]. The microstructural features in a typical ZnO varistor include approximately 90% of a dark grey matrix composed of ZnO grains in the narrow range of few tens of microns [24]. Usually, for this nonlinear, non-ohmic performance, the compositions are doped with a varistor forming oxide (VFO) and primarily involve a $Bi_2O_3$ phase. In order to enhance the threshold voltage and the surge current withstanding capability, the number of intergranular layers in series and parallel, respectively, have to

be enhanced. Therefore, for high surge protection applications, a finer ZnO grain size and a high grain boundary surface area are prerequisites [24–29]. VFO, $Sb_2O_3$, $Cr_2O_3$, $MnO_2/Mn_3O_4$, $SnO_2$, $Al_2O_3$ and $Co_3O_4$ are additionally supplemented in small weight fractions within the composition to act as grain size refiners, inversion boundary (IB) formers, non-linearity factor ($\alpha$) enhancers and the Schottky barrier height promotors [22,26–29], which in a pragmatic sense defines the high voltage transient protection of MOVs [25].

Figure 9 shows the microstructure of the sintered ZnO sample prior to the turning operation. Figure 9a illustrates the top side -1, which was in contact with the open environment (air), depicting a jagged and roughened microstructure due to the low temperature melting behavior and volatility of the Bi-rich phases. Figure 9b shows the surface of the bottom side -2, which remained in contact with the backing ceramic during the sintering and thus the resultant microstructure is much smoother and the secondary phases are uniformly distributed or formed within the matrix. The circumferential cross-section in Figure 9c shows a microstructure similar to the top side which remained in contact with the air during the sintering operation and as a result the morphological features have a coarser surface (although less than the top side). The microstructure of all the sides was also examined by the EDXS quantitative phase evaluation and the cross-section analysis of this region is shown in Table with Figure 9c. Here, the ZnO matrix is prominent, with small weight fractions of $Bi_2O_3$ and $Sb_2O_3$. Other phases are distributed less in the microstructure.

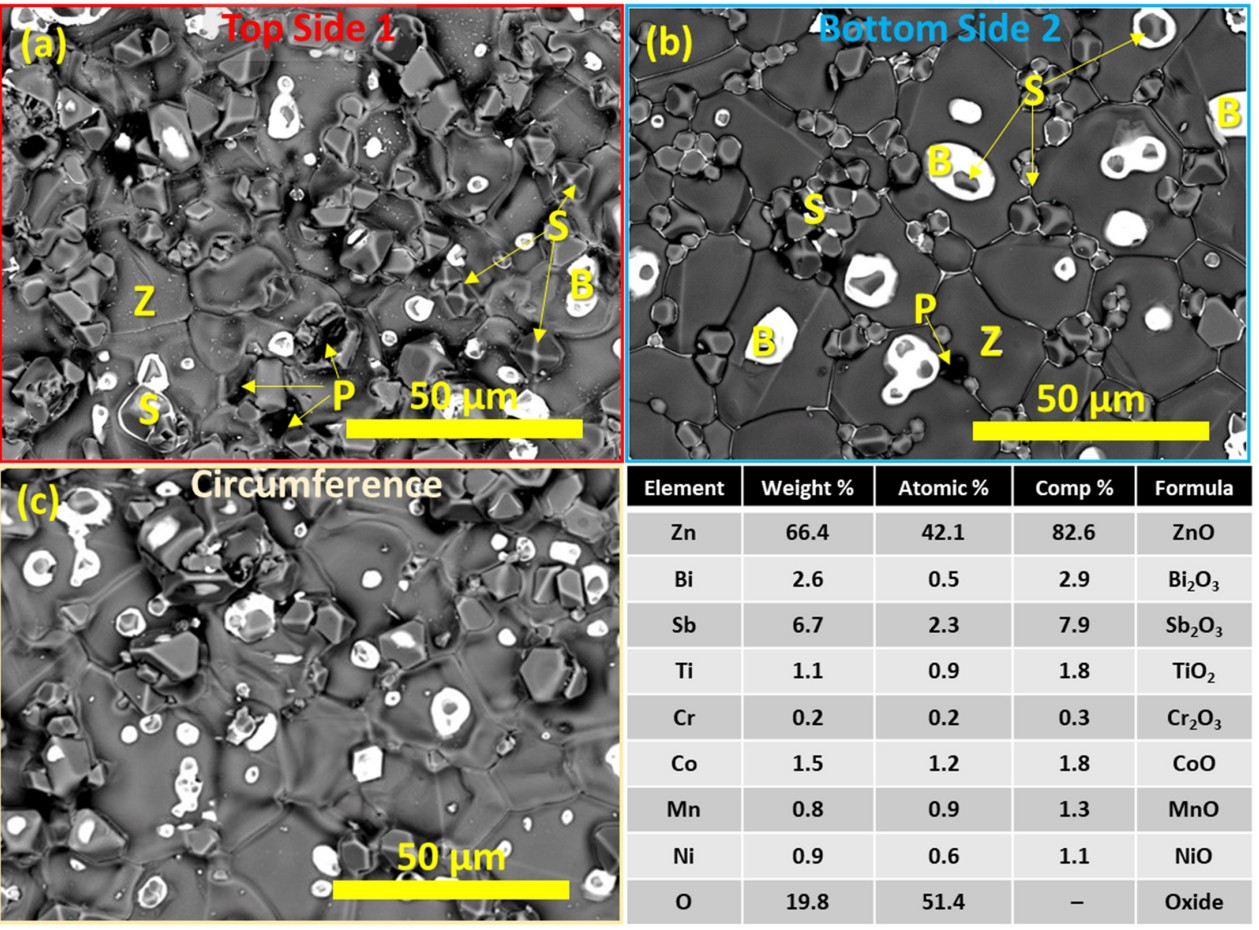

| Element | Weight % | Atomic % | Comp % | Formula |
|---------|----------|----------|--------|---------|
| Zn | 66.4 | 42.1 | 82.6 | ZnO |
| Bi | 2.6 | 0.5 | 2.9 | $Bi_2O_3$ |
| Sb | 6.7 | 2.3 | 7.9 | $Sb_2O_3$ |
| Ti | 1.1 | 0.9 | 1.8 | $TiO_2$ |
| Cr | 0.2 | 0.2 | 0.3 | $Cr_2O_3$ |
| Co | 1.5 | 1.2 | 1.8 | CoO |
| Mn | 0.8 | 0.9 | 1.3 | MnO |
| Ni | 0.9 | 0.6 | 1.1 | NiO |
| O | 19.8 | 51.4 | – | Oxide |

**Figure 9.** Backscattered (BSE) SEM characterization of the ZnO varistor prior to turning, with (**a**) showing the top side of the varistor in contact with air, (**b**) the bottom side 2 of the sample placed on a ceramic boat (microstructure in contact with smooth surface) and (**c**) representing the circumference region of the disk shaped varistor. The table at the end shows the weight, atomic and compound fractions of multiple phases of the EDXS quantified sample in the circumference region.

The $\alpha$-$Bi_2O_3$-rich intergranular phase (IP—defined by 'B' in Figure 9) at the grain boundaries (GBs), triple pockets and ZnO grain junctions appears as a bright constituent in the microstructure with a nominal thickness ranging from 0.1 to 1 µm depending on the sintering conditions [25,26]. Within the composition, $Sb_2O_3$ is added to suppress ZnO grain growth and to enhance the solubility of Zn ions in the $Bi_2O_3$-rich phase at 740 °C eutectic conditions. The presence of $Sb_2O_3$ inclines the transformation thermodynamics towards the $Zn_7Sb_2O_{12}$ spinel phase above 900 °C and this formation is catalyzed by the decomposition of the $Bi_3Zn_3Sb_3O_{14}$ pyrochlore phase above this temperature range during sintering [27,28]. The spinel phase appears as light greyish regions surrounded by the bright $Bi_2O_3$-rich intergranular phase as indicated by the 'S' phase in Figure 9a,b, whereas the pyrochlore phase can be present as scattered whitish precipitates along the grain boundaries and is partially decomposed in the spinel phase. Typically, with Sb/Bi ratio < 1 in the composition, the liquid phase is formed by eutectic melting (ZnO-$Bi_2O_3$) at 740 °C, whereas for Sb/Bi ratio > 1, the $Bi_2O_3$ phase adheres to pyrochlore and the spinel-rich liquid phase appears after decomposition at high temperatures. The pyrochlore decomposition of the spinel and $\alpha$-$Bi_2O_3$ phases leads to a high degree of non-linearity characteristics in MOVs. Typically, without $Sb_2O_3$, the Schottky barrier height drops which increases the donor density of the charges, leading to an increase in the leakage current and a reduction in the non-linearity rated protection. These dopants instinctively reduce the mobility of ZnO-$Bi_2O_3$ GBs due to the pinning effect of the spinel grains and hence enhance the surge protection capabilities [22,24,26,28,29]. With careful EDXS analysis, the microstructure in Figure 9a for the top side and Figure 9c of the cross-section indicated the loss of Bi-rich oxides and a higher Sb/Bi > 1 ratio. More spinel phases were present than IP, as the GBs were running dry in these regions and the resulting average ZnO grain size corresponded to 10–20 µm (top side -1), whereas the matrix grains experienced normal grain growth in the range 15–25 µm (bottom side -2) in the presence of $Bi_2O_3$ IP and liquidus rich GBs. The compound distribution of $Bi_2O_3$ in the central part of the ceramic dropped from ~6% in volume to under ~2.9%, which had a definite impact on the normal grain growth and the secondary phase distribution in the matrix.

Yoshimura et al. [30] scrutinized the mechanical properties of sintered ZnO ceramics and concluded that the samples containing a higher volume fraction of the $Bi_2O_3$-rich phase exhibited lower fracture toughness, whereby the presence of porosity in the sample had little impact towards the fracture toughness. Similarly, they advised that the secondary phases have a low impact on the flexural strength of ZnO ceramics. They further specified from fractography that the cracks initiated from the surface flaws of semi-elliptical shapes and by the Griffith–Irwin relation, the average depth of the critical flaws (c) sized ~50–150 µm (having shape factor 1.13) was approximated for the low relative density samples at 111–137 µm, whereas the higher density samples had a flaw depth estimated at 59–71 µm. The theoretical density of the sintered ZnO specimen accounted for 5.6 g/cm$^3$ and typically the commercially produced samples always comply to >95% relative density. The circular holes in the lower density samples indicate poorly connected granules and can be considered as strength-limiting defects [31]. Nonetheless, the ZnO ceramics exhibit slow crack growth (SCG) susceptibility and usually larger sized ZnO grains prefer trans-granular fracture by crack extension; however, the crack path is invariably intergranular. The presence of secondary phases in higher volume fractions and a finer ZnO grain morphology shift the crack propagation mechanism to an intergranular fracture. Likewise, the hardness of the ZnO ceramics increases with relative compaction nearing the theoretical density up to $H_V$ = 2.3 GPa and this implies microstructural homogeneity with higher secondary phases inducing crack pinning and a reduction in pore volume fraction and size [30].

The grain size has a critical impact on the mechanical properties including cutting as well as the functional properties of surge current/voltage transient resistance. As identified previously [30] and from the work of Ramírez et al. [32], the multiphase ZnO varistors can have a relatively wider size range and matrix grains above 8–20 µm surrounded by a secondary $Bi_2O_3$-rich crystalline liquid phase. The spinel grains and pyrochlore phases

are more prone to cracking than the finer (nanocrystalline) ZnO system. The mechanical cracking in coarser grained ZnO ceramics is propagated through larger Bi-rich spinels and secondary phases due to the ease of the dislocation movement through these softer phases rather than across. Besides, larger grained surfaces during machining are more prone to chipping, preferentially along the edges, causing the formation of pores whose generation is proportional to the feed rates. Larger grained ZnO ceramics preferentially undergo trans-granular fracture by crack extension originating from the surface flaws, including ellipsoidal holes, microcracks and connected porosity from higher feed rates >0.2 mm/rev. Finer sized secondary phases and fewer spinels or pyrochlores in the system should allow crack bridging and pinning, turning the fracture mechanism to intergranular [31].

The accelerometer support in the time domain suggests that this increase in the cutting force exhibited a lower sustainability as the surface roughness was pronounced at the edges and the central part of the sintered ZnO ceramics.

The machining characteristics were analyzed under high resolution scanning electron microscope to corelate the surface topography (in secondary electron—SE mode) after turning with the phase contrast imaging (backscattered electron imaging—BSE) to evaluate the microstructure evolution during the sintering on different sides of green compacts with dissimilar relative densities due to the pressure constriction in uniaxial pressing. The microstructural analysis in Figures 10 and 11 was made at the edge sections (a–f) and the central regions (g–l) of the ZnO varistors.

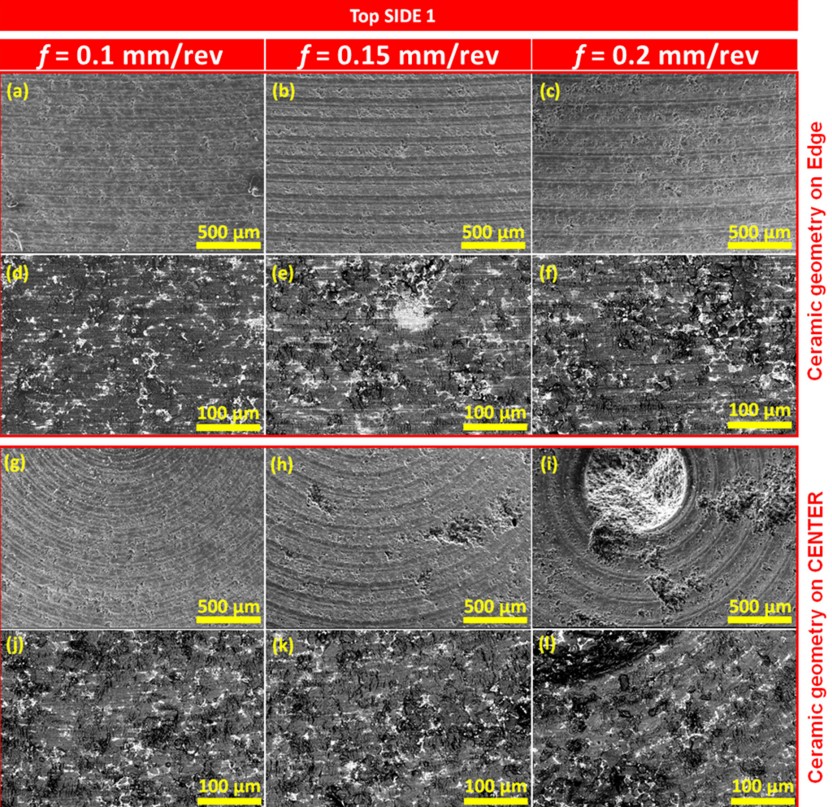

**Figure 10.** SEM analysis of the top side where pressure was applied during green compact processing indicating the variation in the turning characteristics of ZnO varistors for different feed rates (horizontally). SE imaging of the topographical contrast of the surface features after turning in the edges shown from (**a**–**c**) with an increasing feed velocity. BSE phase contrast imaging of the edge region is depicted by (**d**–**f**) with horizontal increases in the feed velocity. The topographical SE imaging in the center of the ZnO varistor is shown for consecutive increases in feed velocities by (**g**–**i**) at $100\times$ magnification, whereas the phase contrast imaging of the central segment of the ceramic is demonstrated from (**j**–**l**) at $500\times$ magnification.

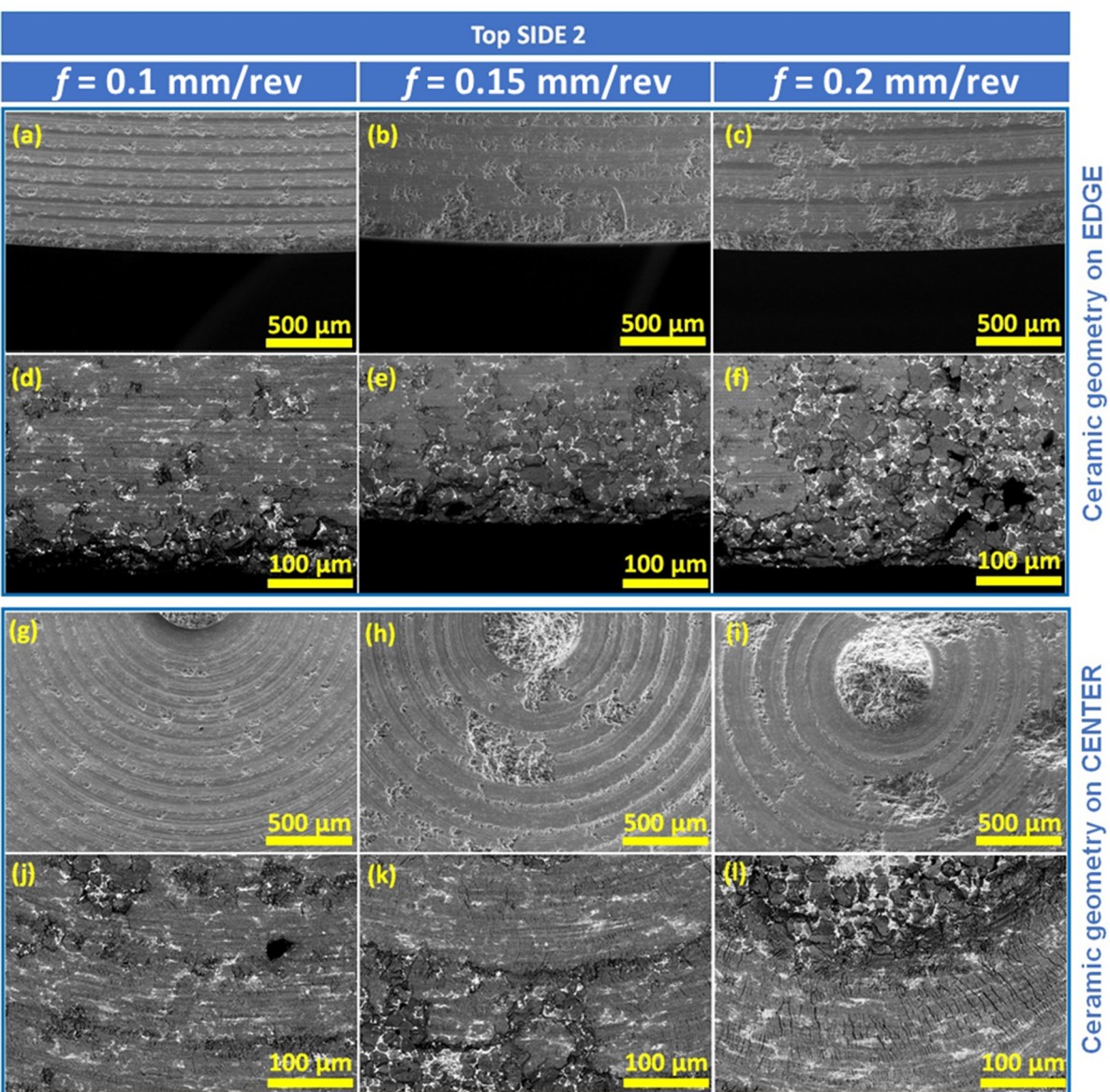

**Figure 11.** SEM analysis of the static bottom side where the uniaxial pressure was not applied during the green compact processing of ZnO varistors representing the variation in the turning characteristics for different feed rates (horizontally). The SE imaging of the topographical contrast of the surface features after turning at the edges is shown from (**a**–**c**) for an incremental feed velocity at $100\times$ magnification. The BSE phase contrast imaging at $500\times$ magnification of the edge region is shown by (**d**–**f**) for a subsequent increase in the feed velocity. The topographical SE imaging at the center of the ZnO varistor is presented for successive surges in feed velocities by (**g**–**i**), while the phase contrast imaging of the central section of the turned ceramic is illustrated from (**j**–**l**).

For these machined electro-ceramics with a high voltage transient protection rating, we anticipated the presence of low faceted spinel phase grains in the vicinity of crystalline $\alpha$-$Bi_2O_3$-rich IP, grain junctions and the GBs along the ZnO matrix [2,26,28]. In Figure 10, the top side of the ceramic where the uniaxial pressure was exerted during the green compaction stage, the ZnO particles were supposed to be distributed differently from the bottom section due to the pressure constraints. The top region was anticipated to redistribute and therefore, the particles are expected to elongate under pressure, but due to the surface in contact with air, the ZnO grains did not experience a similar normal grain growth to the bottom part [33]. As can be seen in Figure 10a–c SE, topographical images of the edge segment of the ceramic with increasing feed velocities show that the machining marks elongated which corresponded to rapid turning action, indicated in

Figure 5 for the 0.2 mm/rev condition. Looking at the BSE images of different turning conditions in the edge region of the top surface of electro-ceramics, the grey ZnO matrix remained in pristine condition after turning at 0.1 mm/rev (Figure 10d). As the feed rate was increased to 0.15 mm/rev and 0.2 mm/rev, the ZnO matrix was subsequently replaced by the darker contrast voids which represent pores due to grain pull-out at higher feed rates. The bright $Bi_2O_3$-rich IP and the spinel phase grains can be seen distributed at triple junction regions and finer ZnO grains. Since this bright phase distribution appears more as a secondary phase, rather than grain boundaries in the machined surface on the top side, this may produce undesirable chipping due to the non-uniform distribution of the eutectic liquid phase during solidification along the ZnO grain boundaries owing to rapid heat exchange at the edges [1]. As the vibrational attenuation in the time and frequency domains suggested that for higher feed rates there were vibrational peaks around 8000 Hz at the edge regions, this must be from chipping which was also verified by the three-dimensional surface topographical analysis shown in the next section. The presence of voids in the edge region only appeared once the feed velocity was increased to 0.2 mm/min, but when we looked at the inside segment of the ceramic, shown in Figure 10g–i, it became quite evident that due to the radial cutting force distribution at the central region, the chipping was pronounced for higher feed rates and thus the resulting surface was rougher. The BSE imaging in Figure 10 j–l shows a similar concentration of bright constituents as secondary phases rather than grain boundaries and the resultant grain pull-out from the central part of ceramic was much more pronounced that the edges, which is visible from the vibrational plots in Figure 6.

For the bottom part of the ZnO varistor, during turning at lower feed velocities the microstructural trend was a reflection of the vibrational feedback from the accelerometer. Conversely, for this case and as the cases shown in Figure 11a–c, we did observe pronounced edge chipping as compared to the top surface in Figure 10a–c, which must have been due to a higher distribution of the spinel phases at the edges. The BSE imaging in Figure 11d–f for the increasing feed velocities confirms that the bright phase distribution was more adequate between the grain boundaries and the secondary phases at the interpunctions or triple pockets; however, the chipping became prominent at the terminal parts of the varistor due to sustained high cutting forces for 0.2 mm/rev. The spinel phases were extremely brittle and were easily ejected from the matrix with a reduction in $Bi_2O_3$ IP towards the edges. Comparing the central segments of the ceramic in the SE topographical imaging, the 0.1 mm/rev showed a lack of surface deterioration, which progressed further as the feed velocity increased up to 0.2 mm/rev, resulting in more pronounced chipping rather than grain pull-out due to the effect of the tangential shear cutting forces [1,3,4]. This roughness of the ceramic edges is yet more protuberant in the phase contrasting images in Figure 11d–f in linear relation with the feed velocities.

Moving to the central part of ceramic, the surface apparent in Figure 11g showed cleaner topographical features at 0.1 mm/rev, whereby a limited grain-pull out was obvious for 0.15 mm/rev in Figure 11h and it extended to surface chipping due to the shear cutting forces at an applied feed rate of 0.2 mm/rev. Therefore, as a customary practice for turning ZnO varistors, we suggest 0.1 mm/rev as the optimal feed velocity for the given cutting parameters: cutting speed ($V_C$) kept at = 100 m/min and spindle rotation speed (n) kept at 3000 rpm. The phase contrast microscopy revealed that the surface area of the secondary phases was relatively higher for the bottom side, which suggests the distribution of a bright $\alpha$-$Bi_2O_3$-rich intergranular phase (IP) at the ZnO–ZnO grain boundaries in Figure 11j–l. Apparently, the chipping-induced fracture topographies on the surface originated from the grain boundary junctions rich with the liquid phase, but due to the brittle nature of the ZnO ceramics, the shear cutting forces induced increased chipping on the surface at 0.2 mm/rev in the central segment, matching the vibrational feedback from the accelerometer. Due to higher shear forces converging at the center, the inside region of the ceramic experienced increased chipping with an increase in the feed velocities, and although the pores were less frequent on this side, the chipping surface fracture was more pronounced in the coarser

grains region than the top side, that contained relatively finer grains which had a high grain pull-out under equivalent turning conditions [1,3]. This work suggests that the spinel grains and the pyrochlore phase create a varistor ceramic more prone to cracking than the finer (nanocrystalline) ZnO system. In the areal comparison, the edge deterioration was still higher with respect to the central part for the bottom side of the machined varistors due to the rapid loss of the Bi-rich phases in a coarser ZnO matrix.

### 3.3. The Three-Dimensional Roughness Measurements of the Machined Surfaces

The roughness measurements with the Alicona three-dimensional microscope are shown in Figure 12 and Figure 14 to expose in detail the topographical variations (surface) and in Figure 12 and Figure 14 for the edge surface (profile) characterization for the respective sides of the sintered ZnO ceramic. Figure 12 refers to the top part of the ceramic which was identified by the electron microscope to exhibit a pronounced grain pull-out at higher feed rates, leading to a rough surface topography. The three-dimensional microscope scanned the regions at the edges to understand the mechanism of material removal with respect to the feed rates [1]. So, for an increase in the feed velocities, the surface roughness by definition of the $S_{10z}$ parameter increased from 17.84 $\mu$m (at 0.1 mm/rev) to 34.71 $\mu$m (0.2 mm/rev). Likewise, the edge radius after the turning operation increased from 65.18 $\mu$m at 0.15 mm/rev to 76.59 $\mu$m at 0.2 mm/rev, which was considerably fine and consistent at 28.65 $\mu$m for 0.1 mm/rev. The three-dimensional optical microscopy images for the edge sections indicated uniformity and a low surface roughness at 0.1 mm/rev which contributed to partial chipping and grain pull-out in the edge of the top surface.

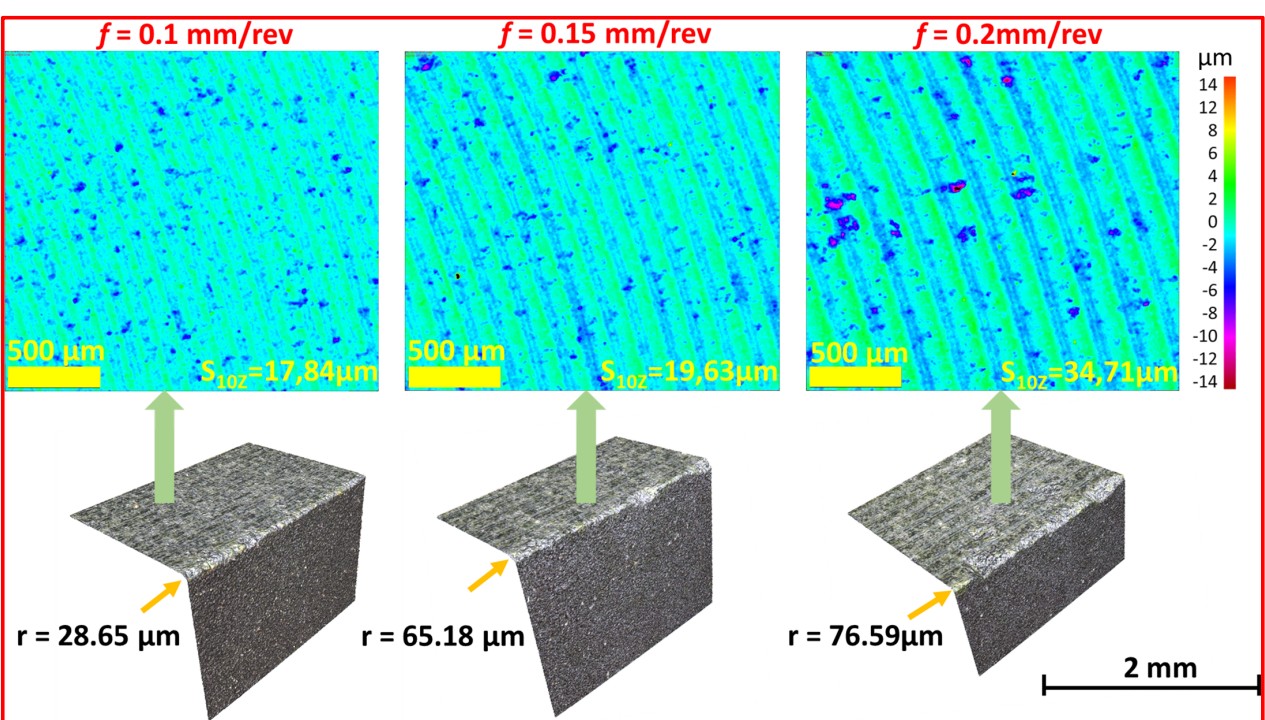

**Figure 12.** The surface roughness profile analyzed for the side 1 i.e., top part of the ceramic.

The profile measurements were also evaluated by 2 × 2 mm area profilograms, as shown in Figure 13 for the top side. The edge roughness parameters such as $R_a$ (average roughness, integral of absolute roughness value), $R_q$ (Root Mean Squared roughness value), and $R_z$ (average peak to valley profile height) were also quantified to validate the turning optimization. The values for $R_a$, $R_q$ and $R_z$ increased linearly with the feed rates from 0.1 mm/rev to 0.2 mm/rev which was in-line with the results obtained from surface roughness measurements in Figure 12. These profile roughness parameters were quite closely spaced for 0.1 and 0.15 mm/rev feed rates. The z-axis or height feedback in

Figure 13 was slightly higher for the rapid feed rates. The profilograms shown in Figure 13 show the widening of the roughness profile for the feed rate 0.2 mm/rev, which implies that a higher degree of surface roughness is imparted if the duration of turning is reduced by one half. Thereby, for an optimal machining scenario and to retain machinability with a diamond PCD cutter up to 1000 passes, the feed rate can be reduced to 0.1 mm/rev.

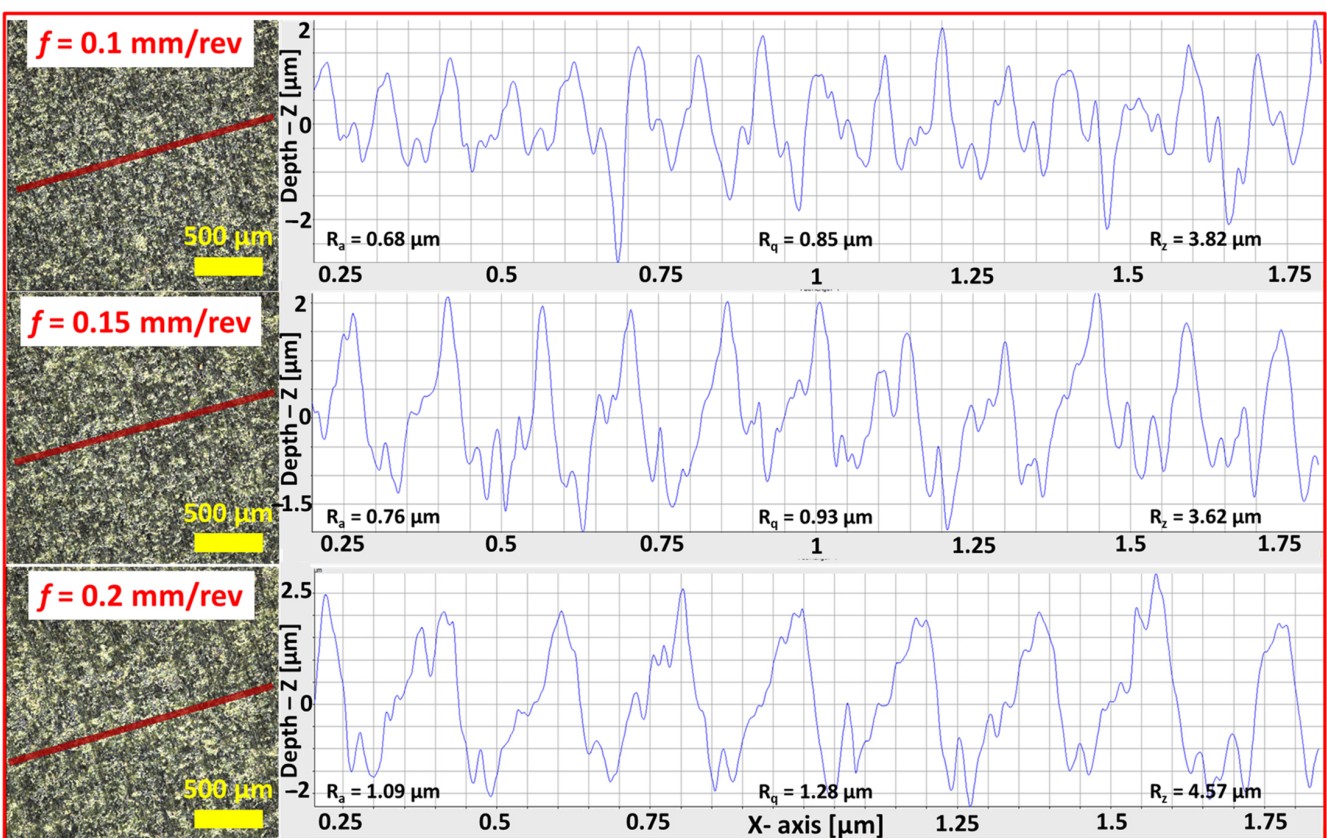

**Figure 13.** The profilograms measuring the variation in the edge roughness parameters for the top side by changing the feed velocities.

Similarly, comparing the bottom surface with the top part of the ceramic, we can deduce that the edge roughness worsened, and the topographical variations became turbulent for the highest 0.2 mm/rev feed velocity. From Figure 14 it is also possible to suggest that the depth of the cut was deeper than in Figure 12; however, at 0.1 mm/rev the surface roughness parameter $S_{10z}$ was measured at 14.96 µm which modestly increased to 22.85 µm for 0.15 mm/rev and successively to 31.21 µm at a 0.2 mm/rev feed velocity. As a comparison, these values for the edges of the bottom part after turning were slightly lower than the circumference measurements of the top surface. The edge radius in Figure 14 for the lowest feed rate was 35.51 µm which increased three-fold to approximately 101.95 µm with 0.2 mm/rev turning conditions. The three-dimensional topographical imaging of the edges suggests that lower feed rates do exhibit surface chipping, which increases their radius in comparison to the top part, in agreement with previous studies [4]. Noticeably, this edge chipping worsened the edge roughness as the radius of the machined edge at 0.15 mm/rev of the bottom half was inferior to the top surface in a 0.2 mm/rev turned ceramic. This effect is related to a combination of edge chipping and grain pull-out on the bottom side of the ceramic as indicated by the topographical color maps in Figure 12, whereas the top surface experienced a more conspicuous grain pull-out which resulted in more uniform edges even at a higher feed velocities [1]. The initial value of R for the pristine non-machined ZnO samples was averaged from 55–109 µm ($S_a$ = 1.37 µm) as shown in Figure 1. With the geometric specifications of the samples and by the optimization of

machining (turning), we were able to reduce the profile roughness and radius R in the range 28.65–35.51 µm ($S_a$ = 0.93 µm), as illustrated in Figures 11 and 13.

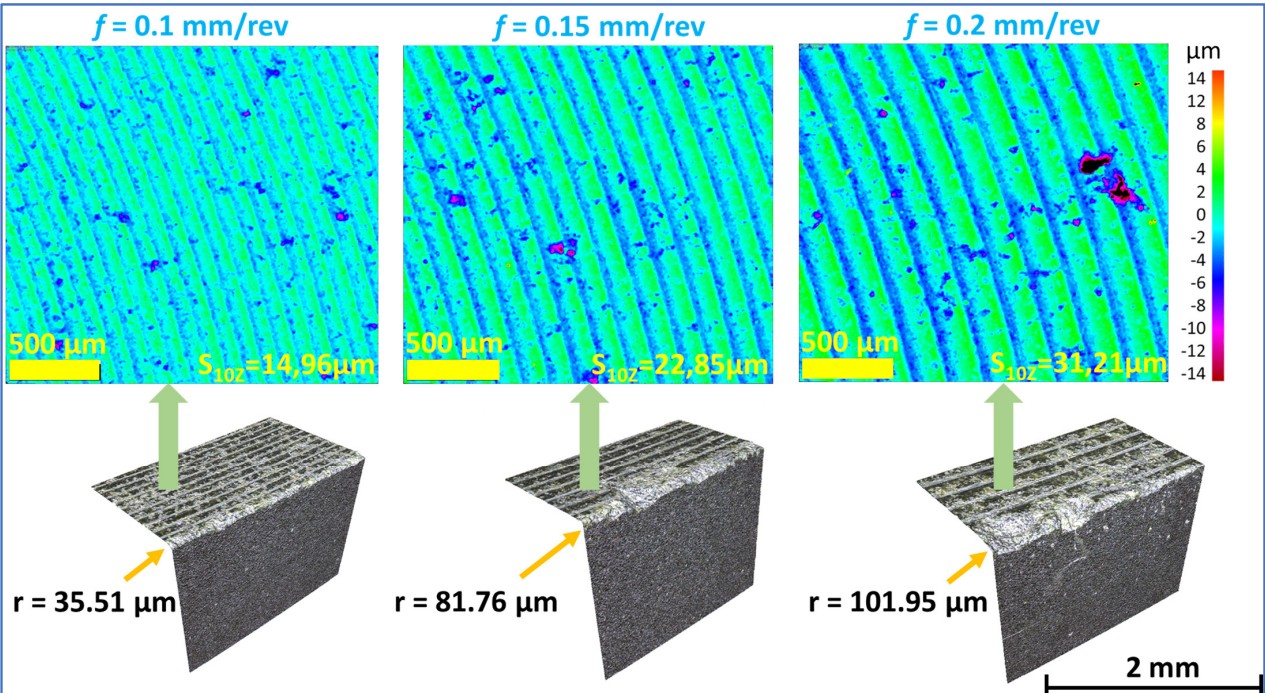

**Figure 14.** The surface roughness profile analyzed for the bottom section i.e., side 2 of the ceramic.

The edge profile for the bottom part was also analyzed to develop a relationship with the surface roughness and the edge profilograms for the varying feed rates are shown in Figure 15. It can be clearly seen that the average profile roughness $R_a$, $R_q$ and $R_z$ of the bottom part was higher than the top side and these values increased simultaneously with the upsurging feed velocities. The average $R_z$ roughness values of the top side at 4.57 µm for 0.2 mm/rev was still marginally lower than that of the bottom part at 5.01 µm for the 0.1 mm/rev setup. The highest obtained $R_z$ = 7.78 µm for 0.2 mm/rev on the bottom side was approximately two-folds higher than the top side and this trend in the roughness profiles was similar for the $R_a$, and $R_q$ values as well. The z-axis peak height and the cutting width separation in the 0.2 mm/rev turning setup as shown in Figure 15 implies that the higher material removal rate was more detrimental than beneficial for the turning optimization and led to a rougher surface over the slower feed rate options.

A comparative edge profile assessment was also carried out on the pristine ceramic before performing the turning operations which can be seen in Figure 16a. The edge radius of the ceramic prior to turning was in the range of 100–125 µm as indicated in Figure 1, whereas the top side of the ceramic measured by the Alicona system shown in Figure 16a had an edge radius of 115.64 µm, rendering the necessity of turning and a high requirement for material removal. Utilizing the most optimal turning setup at 0.1 mm/rev, with the depth of cut ($a_p$) maintained at 0.5 mm and the cutting speed ($V_C$) kept to 100 m/min rendered the final profile radius at 14.22 µm. The corresponding value of the finer edge radius in Figure 16b was even slightly better than the obtained value in Figure 12, but since the ZnO ceramic edge profile sits in a dormant range, a tight geometric control is still possible with the conventional machining and turning setups as explained in this study.

Therefore, in light of our findings, this report suggests that the vibrational attenuation obtained from the accelerometer suggests that the surface roughness of microstructures at high feed velocities and in the case of a sintered ZnO ceramic, the conventional turning and milling operations should be performed considering edge chipping as a sustained failure mechanism but optimizing the feed rates and the depth of cut [1]. Higher tangential

shearing forces result in extremely rough edges which make it hard to apply dielectric epoxy or glass layers to insulate the varistor in electrical assembly. Moreover, this high surface roughness in the center faces results in a bad deposition of the metallization layers and as a result, these high current grade surge protectors typically fail impulse tests during quality control rather than going forward with the practical application [25].

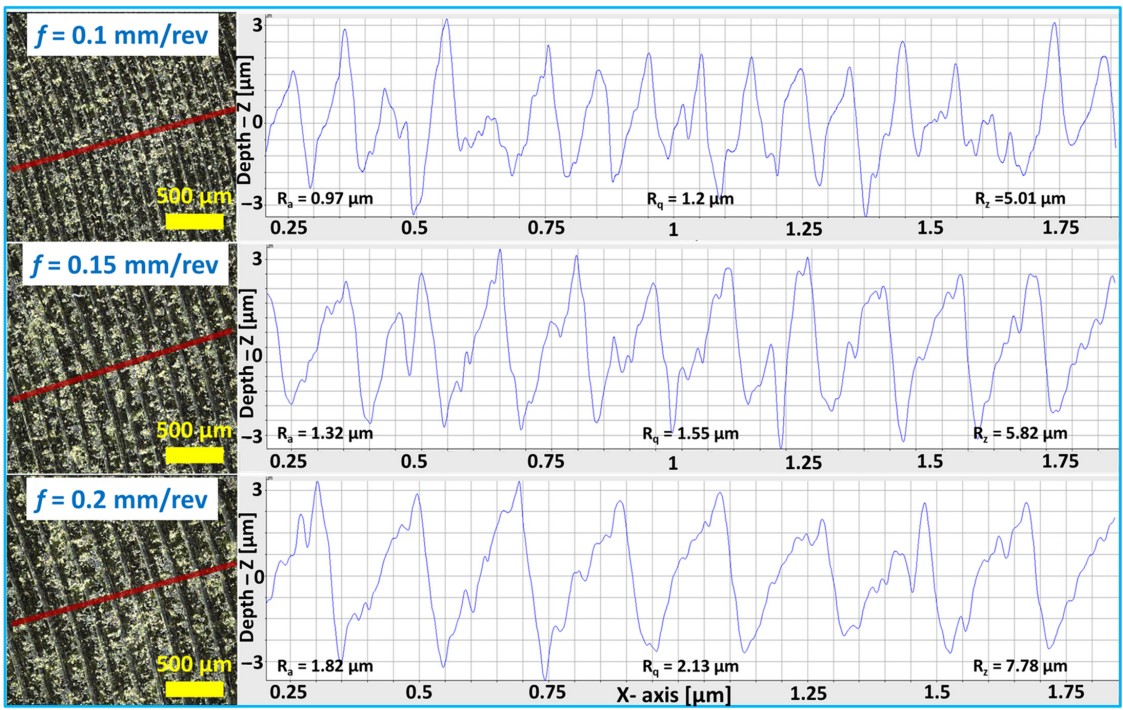

**Figure 15.** The edge roughness profilograms analyzed for the bottom section i.e., side 2 of the ZnO ceramic for different feed rates during turning.

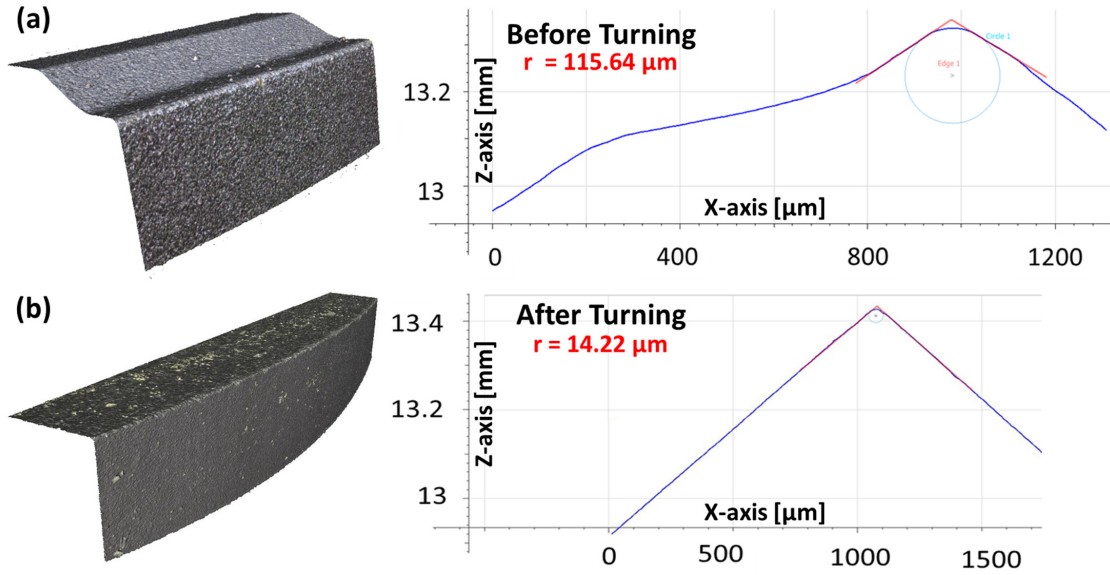

**Figure 16.** The edge profile comparison between the (**a**) non-machined and (**b**) optimally turned ZnO ceramic.

Typically, the failure during electrical impulse testing is directly related to the thermomechanical properties, microstructure and composition, rather than the pulse duration

and thus the coarser grained ZnO varistors are only applicable for low voltage transient protection applications [32].

## 4. Conclusions

In this short study on the application of an accelerometer for returning feedback related to the turning optimization of ZnO electro-ceramics, we reported the controlling feed rates to obtain uniform and chip free surfaces at 0.1 mm/rev with a cutting speed ($V_C$) = 100 m/min and a spindle rotation speed (n) = 3000 rpm. For high feed velocities, although the turning operation was completed in a shorter time domain, the very high vibrational amplitude suggested excessive cutting forces which in turn translated to high edge roughness and pronounced chipping at 0.2 mm/rev. The machining optimization was closely connected with the microstructural features, thus suggesting that the edge chipping was higher on the side where the microstructure was more uniform, whereas the grain pull-out became noticeable when the distribution of the secondary phases was higher in the matrix. The topographical color maps indicated that the surface and edge roughness increased as the feed rate became unsustainably higher and the larger shear forces should effectively render these functional ceramics useless for high transient or surge protection applications as the surface metallization and circumferential insulation will be non-uniformly deposited on the varistor.

**Author Contributions:** Conceptualization, F.P.; methodology, F.P.; software, A.I. and F.P.; validation, F.P. and A.I.; formal analysis, J.D.; investigation, J.D.; resources, F.P. and J.D.; data curation, F.P. and A.I.; writing—original draft preparation, A.I.; writing—review and editing, A.I. and J.D.; visualization, F.P., A.I. and J.D.; supervision, F.P.; project administration, F.P.; funding acquisition, F.P. All authors have read and agreed to the published version of the manuscript.

**Funding:** This research work received financial support from the Slovenian Research Agency under the project serial numbers: L2-8184 and L2-1836.

**Institutional Review Board Statement:** Not applicable.

**Informed Consent Statement:** Not applicable.

**Data Availability Statement:** Not applicable.

**Acknowledgments:** We would also like to thank Slavko Bernik from the Department for Nanostructured Materials at the Jožef Stefan Institute Slovenia for his support in performing the scanning electron microscopy analysis on the machined ZnO ceramics.

**Conflicts of Interest:** The authors declare no conflict of interest.

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
