# Peer review of "Evaluation of Chip Formation Mechanisms in the Turning of Sintered ZnO Electro-Ceramics"

_processes, doi:10.3390/pr9081422_

Round 1

Reviewer 1 Report

The manuscript has been well written and needs a minor revision before acceptance. 

  1. How about the damage depth beneath the surface after cutting using different speed?
  2. what is the damage mode of brittle ZnO grains during the cutting?
  3. Whether the grain size of ZnO has affected the surface quality when the author modified the feed velocity? 
  4. How to remove the microcracks finally?

Author Response

Reviewer #1:

The manuscript has been well written and needs a minor revision before acceptance. 

Thank you for the review and returning suitable comments to improve the lucidity of scientific/engineering work. All the questions posted by the reviewer has been answered in this letter.

  1. How about the damage depth beneath the surface after cutting using different speed?

Thank you for this query.

The damage depth analysis is usually done by fractography whereas in our study we performed surface characterization of turned ceramics to preclude the formation of surface defects e.g., pores and chipping in relation to the ZnO grain size, secondary phases, and processing. The identification of the average depth of critical flaws (c) in ZnO sintered ceramics has been classified in the size range of 50 – 150 µm with the shape factor 1.13 for the samples with low relative density samples at 111 – 137 µm by Yoshimura et al. [1]. They further suggested that the samples with higher relative density had flaw depth reduced to 59 – 71 µm. Since these samples under investigation are above 95% compact according to industrial quality control, we anticipate average depth of critical flaws around 50 - 100 µm, however such an identification was out of scope of this study. The radius (r) values of the edges in both sides is less than 130 µm. With the increase in cutting speed, it can be anticipated that the increase vibrations should have drastic impact on the sample and thus from the microstructure of both sides, we can clearly see the r values increasing above 100 µm. The damage depth should increase substantially with feed rates above 0.2 mm/rev as the pore size in general can be approximated from SEM images in the range of 150 – 250 µm, thus the damage depth which can only be classified by fractography should be significantly higher than at 0.1 mm/rev.

The related details in the manuscript for citing the relationship of cutting and material removal mechanism with the mechanical properties is shown from line 323 - 357:

Yoshimura et al. [30] scrutinized the mechanical properties of sintered ZnO ceramics and concluded that the samples containing higher volume fraction of Bi2O3-rich phase exhibit lower fracture toughness, whereby the presence of porosity in the sample had little impact towards the fracture toughness. Similarly, they advised that the secondary phases have low impact on the flexural strength of ZnO ceramics. They further specified from fractography that the cracks initiated from the surface flaws of semi-elliptical shape and by Griffith-Irwin relation, the average depth of critical flaws (c) sized ~50 – 150 µm (having shape factor 1.13) was approximated for low relative density samples at 111 – 137 µm, whereas the higher density samples had flaw depth estimated at 59 – 71 µm. The theoretical density of sintered ZnO specimen account for 5.6 g/cm3, and typically the commercially produced samples always comply to >95% relative density. The circular holes in lower density samples indicate poorly connected granules and can be considered as strength-limiting defects [31]. Nonetheless, the ZnO ceramics exhibit slow crack growth (SCG) susceptibility and usually larger sized ZnO grains prefer transgranular fracture by crack extension, however, the crack path is invariably intergranular. The presence of secondary phases in higher volume fraction and finer ZnO grain morphology shift the crack propagation mechanism to intergranular fracture. Likewise, the hardness of ZnO ceramics increase with relative compaction nearing theoretical density up to HV = 2.3 GPa and this implies microstructural homogeneity with higher secondary phases inducing crack pinning and by reduction in pore volume fraction and size [30].

The grain size has critical impact on the mechanical properties including cutting as well as the functional properties of surge current/voltage transient resistance. As identified previously [30] and from the work of Ramírez et al. [32], the multiphase ZnO varistors can have a relatively wider size range and matrix grains above 8 – 20 µm surrounded by secondary Bi2O3-rich crystalline liquid phase, the spinel grains and pyrochlore phase are more prone to cracking than finer (nanocrystalline) ZnO system. The mechanical cracking in coarser grained ZnO ceramics propagate through larger Bi-rich spinels and secondary phases due to ease of dislocation movement through these softer phases rather across. Be-sides larger grained surfaces during machining are more prone to chipping, preferentially along the edges causing formation of pores whose generation is proportional to the feed rates. Larger grained ZnO ceramics preferentially undergo transgranular fracture by crack extension originating from the surface flaws, including ellipsoidal holes, microcracks, connected porosity from higher feed rates > 0.2 mm/rev. Finer sized secondary phases and less spinels or pyrochlores in the system should allow crack bridging and pinning, turning fracture mechanism to intergranular [31].”

  1. what is the damage mode of brittle ZnO grains during the cutting?

Since the cutting operations were done without any liquid medium, we can rule out the possibility of brittle fracture inhibition by hydrostatic pressure generated at the contact surfaces leading to plasticity induced fracture, and thus in practice during dry cutting the damage mode in sintered ZnO ceramics should be rather confined to brittle fracture mode as no wear (including tribochemical) or erosion is expected either.

Yoshimura et al. [1] studied the mechanical properties of sintered ZnO ceramics and deduced that the samples containing higher volume fraction of Bi2O3-rich phase exhibit lower fracture toughness and the presence of porosity in the sample had little impact towards the fracture toughness. Similarly, they suggested that the secondary phases have low impact on the flexural strength of ZnO ceramics. They further detailed from fractography that the cracks initiated from the surface flaws of semi-elliptical shape and by Griffith-Irwin relation, the average depth of critical flaws (c) sized ~50 – 150 µm (having shape factor 1.13) was approximated for low relative density samples at 111 – 137 µm, whereas the higher density samples had flaw depth estimated at 59 – 71 µm. The theoretical density of sintered ZnO specimen account for 5.6 g/cm3, whereby the commercially produced samples always comply to >95% relative density. The circular holes in lower density samples indicate poorly connected granules and can be considered as strength-limiting defects [2]. Nonetheless, ZnO ceramics exhibit slow crack growth (SCG) susceptibility and usually larger sized ZnO grains prefer transgranular fracture by crack extension, however, the crack path is invariably intergranular. The presence of secondary phases in higher volume fraction and finer ZnO grain morphology shift the crack propagation mechanism to intergranular fracture. Likewise, the hardness of ZnO ceramics increase with relative compaction nearing theoretical density up to HV = 2.3 GPa and this implies microstructural homogeneity with higher secondary phases inducing crack pinning and by reduction in pore volume fraction and size [1].

  1. Whether the grain size of ZnO has affected the surface quality when the author modified the feed velocity? 

Inarguably, this is correct anticipation. Yes, the grain size has critical impact on the mechanical properties including cutting as well as the functional properties of surge current/voltage transient resistance. As mentioned before [1] and from the work of Ramírez et al. [3] that the multiphase ZnO varistors can have a relatively wider size range and matrix grains above 8 – 10 µm surrounded by secondary Bi2O3-rich crystalline liquid phase, the spinel grains and pyrochlore phase are more prone to cracking than finer (nanocrystalline) ZnO system. The mechanical cracking in coarser grained ZnO ceramics propagate through larger Bi-rich spinels and secondary phases due to ease of dislocation movement through these softer phases rather across. Besides larger grained surfaces during machining are more prone to chipping, preferentially along the edges causing formation of pores whose generation is proportional to the feed rates. Larger grained ZnO ceramics preferentially undergo transgranular fracture by crack extension originating from the surface flaws, including ellipsoidal holes, microcracks, connected porosity from higher feed rates > 0.2 mm/rev. Finer sized secondary phases and less spinels or pyrochlores in the system should allow crack bridging and pinning, turning fracture mechanism to intergranular [2].

Failure during electrical impulse testing is directly related to the thermomechanical properties, microstructu8re and composition, rather than the pulse duration and thus the coarser grained ZnO varistors are only applicable for low voltage transient protection applications [3].

These details are also inscribed in the manuscript as an answer to question number 1.

  1. How to remove the microcracks finally?

This is a very nice argument. The microcracks essentially are not considered a problem in multi-phase ZnO ceramics as long as the relative density of the sample is higher than 95%. However, larger sized cracks act as barrier to the flow of electric current and they reduce the functional stability of the varistors.

The damage depth analysis can be done by fractography, whereas we performed surface and profile characterization of turned ceramics to determine the causes leading to the formation of surface defects e.g., pores and chipping in relation to the ZnO grain size, secondary phases, and processing. The identification of the average depth of critical flaws (c) in ZnO sintered ceramics has been classified in the size range of 50 – 150 µm with the shape factor 1.13 for the samples with low relative density samples at 111 – 137 µm by Yoshimura et al. [1]. They further suggested that the samples with higher relative density had flaw depth reduced to 59 – 71 µm. Since these samples under investigation are above 95% compact according to industrial quality control, we anticipate average depth of critical flaws around 50 - 100 µm, however such an identification was out of scope of this study. The radius (r) values of the edges in both sides is less than 130 µm. With the increase in cutting speed, it can be anticipated that the increase vibrations should have drastic impact on the sample and thus from the microstructure of both sides, we can clearly see the R values increasing above 100 µm. The damage depth should increase substantially with feed rates above 0.2 mm/rev as the pore size in general can be approximated from SEM images in the range of 150 – 250 µm, thus the damage depth which can only be classified by fractography should be significantly higher than at 0.1 mm/rev.

One method to control microcracking (including sub-surface cracking) employs metallization thermal treatment at lower temperature than sintering to allow redistribution of Bi-rich and spinel phases at grain boundaries and internal/surface defects including triple pocket regions. Presence of secondary phases contribute to microcrack and dislocation pinning at the jagged interfaces which increases the elastic modulus of the material and that is the reason why crack propagation nominally follows intergranular route. Reducing the number of faceted spinel phases (shown in Figure 9) and open pores by making composition Sb/Bi ratio < 1 should favor an interfacially Bi2O3-rich microstructure in ZnO matrix, resulting in an improvement of fracture toughness and elastic modulus.

However, microcracks formed due to shear forces may tend to elongate upon sustained loads, which can be mitigated by reducing the feed rates, depth of cut, cutting tool edge angle and partially heating the workpiece to counter the effect of tangential shear forces [1, 4].

Therefore, two methods for industrial practice we can suggest, the first one is post-sinter finishing and utilizing micro-polishing or honing to remove these microcracks after the workpieces have been optimally machined, whereas the second practice should be optimization of composition and processing profiles such that the microstructural homogeneity comes built-in after sintering. In this way, the machining/turning is necessary in case of low voltage transient protection varistors only, since they may randomly contain holes, connected porosity and microcracks due to grain coarsening in (thermally) prestressed ceramics. Micro-polishing should be the most optimum method to treat these surface heterogeneities in the coarser grained ZnO ceramics, whereas microstructural control in finer (nanocrystalline) ZnO varistors should intrinsically circumvent surface defects volume fraction and the need to prepare the surfaces by machining.

At the end, we sincerely thank the reviewer for spending generous time to review the manuscript and provide us with practical feedback which was prerequisite for successfully publishing with MDPI Processes.  In case you may find additional segments to correct or comments/advice, we would be obliged to facilitate you without hesitation.

References (in this letter of revision)

[1]          H. N. Yoshimura, A. L. Molisani, N. E. Narita, J. L. A. Manholetti, and J. M. Cavenaghi, "Mechanical Properties and Microstructure of Zinc Oxide Varistor Ceramics," Materials Science Forum, vol. 530-531, pp. 408-413, 2006.

[2]          B. Balzer, M. Hagemeister, P. Kocher, and L. J. Gauckler, "Mechanical Strength and Microstructure of Zinc Oxide Varistor Ceramics," Journal of the American Ceramic Society, vol. 87, no. 10, pp. 1932-1938, 2005.

[3]          M. A. Ramírez et al., "Mechanical Properties and Dimensional Effects of ZnO- and SnO2-Based Varistors," Journal of the American Ceramic Society, vol. 91, no. 9, pp. 3105-3108, 2008.

[4]          D. Muženič, J. Dugar, D. Kramar, M. Jezeršek, and F. Pušavec, "Improvements in Machinability of Zinc Oxide Ceramics by Laser-Assisted Milling," Strojniški vestnik – Journal of Mechanical Engineering, no. 10, pp. 539-546, 2019.

Reviewer 2 Report

No comments

Author Response

We genuinely thank the reviewer for devoting their substantial time and effort to review the manuscript and highlighting its suitability for publishing with MDPI Processes.

Reviewer 3 Report

This submission describes results of the evaluation of chip formation mechanisms in turning of sintered ZnO electro-ceramics. This subject is interesting; however, some remarks should be done:

  1. Detailed information about structures, hardness, porosity and so on of both sides of the samples should be added,
  2. Detailed information on cutting edge angles and full insert designation should be added in the text,
  3. What does it mean [g] measuring unit? Is real acceleration was equal –50 g, as in Fig. 5 and 7?
  4. Measuring units should be added in Fig. 6 and 8,
  5. Only SEM images have been shown in Fig. 9, not SEM and microstructural analysis results; it should be presented,
  6. Information about ceramics properties (see remark 1) should be used to explain cutting features,
  7. Surface images on the Fig. 11, 12 are completely uninformative, it would be better to show 2D profilograms,
  8. What method was used to measure the edge radius with an accuracy of two decimal places? And what was the value of this radius for the initial edge?
  9. English should be improved and numerous typos should be corrected.

Author Response

Please find answers to relevant reviewer queries in the attachment.

Kind Regards

Reviewer 4 Report

I do not find some results about chip formation, which presents in the title. The scientific undestanding of cutting processes could not sound in this article. The article looks like an experimental report. The relationship between signals obtained during the cutting processes and parameters measured after the cutting processes should be built in a scientific way.

There are some specific comments:

  1. What is the runout of the ZnO sample during its rotation? Not only the cutting tool vibration, but also the runout of the ZnO sample lead to the chipping of the ZnO sample.
  2. The chipping occurs on the edges of ZnO sample. If the chamfering process is performed before the turning process, the chipping of ZnO sample should be reduced.
  3. Turning process could induce the micro-crack in the subsurface of ZnO sample. Please present how to control the surface integrity.
  4. The tool wear could be considered in the turning process of hard and brittle materials. Please provide more details for the wear of cutting tools.
  5. "chip formation" and "chipping formation" is totally different concept. Please check them.

Author Response

(The authors gave the same response as above.)

Round 2

Reviewer 3 Report

I agree with changes made